# Evaluation of the EarthCARE Cloud Profiling Radar (CPR) Doppler velocity measurements using surface-based observations

Jiseob Kim<sup>1</sup>, Pavlos Kollias<sup>1,2</sup>, Bernat Puigdomènech Treserras<sup>1</sup>, Alessandro Battaglia<sup>3</sup>, Ivy Tan<sup>1,4</sup>

- <sup>1</sup>Department of Atmospheric and Oceanic Sciences, McGill University, Montreal, Canada
- <sup>2</sup>School of Marine and Atmospheric Sciences, Stony Brook University, Stony Brook, NY, USA
  - <sup>3</sup>Department of Environment, Land and Infrastructure Engineering (DIATI), Politecnico of Torino, Torino, Italy
  - <sup>4</sup>Department of Physics, University of Colorado Boulder, Boulder, CO, USA

Correspondence to: Jiseob Kim (jiseob.kim@mcgill.ca)

Abstract. The Earth Cloud, Aerosol and Radiation Explorer (EarthCARE) mission was launched on May 28, 2024. One of the most exciting new measurement capabilities of the EarthCARE mission is the CPR Doppler velocity measurement. The availability of Doppler measurements from space will offer a unique opportunity for the collection of a global dataset of vertical motions in clouds and precipitation. An important step in realizing this opportunity is to evaluate the CPR Doppler velocity measurements against those collected by surface-based observatories. Validation with two high-latitude surface-based Doppler radar observatories demonstrates that the CPR Level-2 Doppler velocities exhibit minimal biases (within a few cm/s), especially in ice clouds. Even in low-level mixed-phase clouds, the CPR's Doppler velocity measurements provide reliable values, although careful consideration is needed for specific limitations such as vertical smoothing effects due to the radar's pulse length. Despite the inherent challenges associated with space-based Doppler measurements, these results suggest strong potential for the EarthCARE mission to provide unprecedented global climatological insights into hydrometeor sedimentation velocities.

#### 20 1 Introduction

Hydrometeor terminal fall velocity is fundamental for numerous processes within clouds, and hence should be accurately represented in weather and climate models. The hydrometeor fall velocity modulates the vertical transport of hydrometeors, cloud phase partitioning, cloud lifetime, and precipitation efficiency (Blyth et al., 2005; Mitchell and Finnegan, 2009; Heymsfield and Westbrook, 2010; Donner et al., 2016; Tan and Storelvmo, 2016). This is particularly true for the terminal velocity of solid hydrometeors (ice and snow). Ice particle fall velocity represents a key factor for numerous important processes in the atmosphere such as the vertical redistribution of water vapor (D'Alessandro et al., 2019), latent and radiative heating and cooling within clouds (Nelson and L'Ecuyer, 2018), and microphysical evolution of precipitation including hail formation, aggregation, riming, and sublimation (Heymsfield et al., 1980; Chase et al., 2018; Shates et al., 2021). In addition, it contributes to cloud feedbacks that influence climate sensitivity, atmospheric circulation patterns, radiative balance, and the surface energy budget (Bony et al., 2015; Tan et al., 2016; Hofer et al., 2019).

Until now, the terminal velocity of hydrometeors has been estimated using theoretical calculations or observations. Fall velocities of frozen hydrometeors of different habits have been estimated using relationships between particle mass and projected area as a function of particle diameter (e.g., Mitchell, 1996; Heymsfield and Westbrook, 2010; Szyrmer et al., 2012) and in situ particle sampling sensors (Vázquez-Martín et al., 2021). Surface-based profiling Doppler radars have provided valuable information about the hydrometeor terminal velocity, often expressed as functions of the mean Doppler velocity and their radar reflectivity (Protat and Williams, 2011; Kalesse and Kollias, 2013; Kalesse et al., 2013; Matrosov, 2023).

The Earth Cloud, Aerosol and Radiation Explorer (EarthCARE; Illingworth et al., 2015; Wehr et al., 2023) satellite was launched on May 28, 2024. One of the core instruments of the EarthCARE satellite is a 94 GHz Cloud Profiling Radar (CPR) with Doppler velocity measurement capability (Kollias et al., 2018, 2022). The EarthCARE CPR has higher sensitivity, better horizontal resolution, and reduced surface clutter contamination (Illingworth et al., 2015; Burns et al., 2016; Lamer et al., 2020) compared to the National Aeronautics and Space Administration (NASA) CloudSat CPR (Tanelli et al., 2008; Stephens et al., 2008, 2018). Moreover, the EarthCARE CPR's Doppler velocity measurements can provide the first global climatology of hydrometeor terminal velocity using techniques similar to those employed by surface-based radars.

Measuring Doppler velocity from space, however, presents unique technical challenges due to the satellite's high platform velocity (7.6 km s<sup>-1</sup>). The high relative speed of an orbiting radar with respect to the hydrometeors introduces significant broadening in the Doppler spectrum (Kobayashi, 2002; Tanelli et al., 2005) that increases the Doppler velocity measurement uncertainty and bias (Kollias et al., 2014). This effect is particularly pronounced under low signal-to-noise ratio (SNR) and non-uniform beam filling (NUBF) conditions (Illingworth et al., 2015; Battaglia et al., 2020; Kollias et al., 2022). In addition, errors due to uncertainty in the CPR antenna pointing characterization can lead to Doppler velocity biases (Tanelli et al., 2005; Battaglia and Kollias, 2014). To mitigate these issues, European Space Agency's (ESA's) CPR Level-2 (L2) processing algorithms apply sophisticated corrections (e.g., NUBF corrections, spatial averaging, and mispointing adjustments), ultimately providing quality-controlled Doppler velocity best estimates (as detailed by Kollias et al., 2023; Puigdomènech et al., 2025).

The objective of this study is to validate the post-launch performance of the CPR Doppler velocity measurements using surface-based Doppler radars, thereby confirming their reliability before they are applied in research for precipitation process understanding or for the estimation of a global climatology of hydrometeor terminal velocity. Observations from two high-latitude surface-based Doppler radar observatories are utilized. A CPR instrument simulator is applied to the surface radar observations to minimize the differences in measurement characteristics between space- and surface-based radars (Pfitzenmaier et al., 2025). This allows us to transform surface-based observations into CPR-like synthetic data, making them more comparable. Details on the satellite and surface-based radar datasets, as well as the instrument simulator, are provided in Section 2. In Section 3, we first validate the CPR Doppler velocities in ice clouds, and then assess their performance and limitations in low-level mixed-phase clouds. Section 4 concludes with a summary of key findings.

## 2 Data and methodology

Here, the methodology applied to compare the EarthCARE CPR L2 Doppler velocity and the surface-based Doppler velocity measurements is described. The instrument simulator is introduced alongside the reference dataset used for validation, and the processing steps applied to ensure consistency between spaceborne and surface-based observations are outlined.

### 2.1 Estimation of sedimentation velocity from Doppler velocity

Profiling Doppler radars measure a Doppler velocity  $(V_D)$  that represents the sum of the reflectivity-weighted hydrometeor fall velocity  $(V_F)$  and the vertical air motion  $(V_A)$ :

$$V_D = V_F + V_A \ . \tag{1}$$

The estimation of  $V_F$  from  $V_D$  requires the removal of the vertical air motion contribution (Kollias et al., 2002). Several methods have been proposed to estimate  $V_F$  from  $V_D$  (Orr and Kropfli, 1999; Matrosov et al., 2002; Protat et al., 2003; Delanoë et al., 2007; Plana-Fattori et al., 2010). The main assumption is that, by averaging over a sufficiently large time–space window, the mean vertical air motion becomes zero ( $V_A \approx 0$ ). A commonly used technique for this approach is the  $V_F - Z_e$  method, which employs an empirical power-law relationship between radar reflectivity ( $Z_e$ , in mm<sup>6</sup> m<sup>-3</sup>) and fall velocity:

$$V_D = a Z_e^b \,, \tag{2}$$

where the coefficients a and b are determined via least squares regression (Protat et al., 2003; Kalesse and Kollias, 2013; Matrosov, 2023). An extended version of this approach, often called  $V_F - Z_e - H$  (Plana-Fattori et al., 2010; Protat and Williams, 2011), adds height (H) as an extra parameter to capture the fact that ice particle habit and density can change with altitude, reflecting changes in temperatures, and thus alter the relationship between reflectivity and fall velocity.

## 2.2 EarthCARE CPR Doppler velocity

Here, the post-processing of the raw CPR Doppler velocity measurements is described. First, the CPR antenna mispointing is characterized and the associated Doppler velocity bias is removed using a dedicated L2 processor called the CPR antenna pointing characterization (C-APC), which uses satellite attitude and orbit control system data, along with additional references (e.g., ocean surfaces, snow-covered land, or ice clouds), to quantify and remove residual mispointing (Tanelli et al., 2005; Battaglia and Kollias, 2014). The details of the CPR antenna pointing correction and its evaluation are described in Puigdomènech et al. (2025). After the mispointing corrections are applied, the CPR corrected Doppler (C-CD) algorithm addresses residual issues such as NUBF and velocity folding (for more details, see Kollias et al., 2023).

The corrections are applied at the 1 km along-track resolution. The CPR Doppler velocity estimates are noisy due to uncertainty introduced by the platform motion. Therefore, the C-CD algorithm applies spatial averaging with a window of 4 km in the horizontal and 500 m in the vertical to reduce Doppler velocity uncertainty. Kollias et al. (2023) demonstrated that such averaging could reduce the uncertainty from about 1.5 m s<sup>-1</sup> to 0.5 m s<sup>-1</sup>, although they used a slightly different averaging window of 5 km horizontally and 300 m vertically. Notably, this spatial averaging is applied to the real and

imaginary components of the complex covariance obtained from the pulse pair processing, rather than to Doppler velocity values directly.

## 2.2.1 Sedimentation velocity best estimate (SVBE)

Building on the corrected Doppler data, the C-CD algorithm estimates the reflectivity-weighted hydrometeor fall speed. This procedure, similar to the  $V_F - Z_e - H$  method (Plana-Fattori et al., 2010; Protat and Williams, 2011; Kalesse and Kollias, 2013), involves binning the Doppler velocities by radar reflectivity within a narrow vertical window of 300 m, and four along-track windows (40, 30, 20, and 10 km). To ensure statistical robustness, each bin must contain at least five valid Doppler values. If this criterion is not met at smaller windows, values from the next larger window are used instead. Under the assumption that reflectivity and vertical air motion are uncorrelated on these scales, the mean Doppler velocity within each bin converges to the sedimentation velocity (Illingworth et al., 2015; Kollias et al., 2022). The retrieval is generally not performed below -21 dBZ because the SNR falls below acceptable limits (Kollias et al., 2022; Puigdomènech et al., 2025). By combining these binning and spatial-averaging procedures, uncertainties in the SVBE can be lowered to about 0.3 - 0.4 m s<sup>-1</sup> under ice-cloud and light-precipitation conditions (Kollias et al., 2023).

In this study, validation is conducted only from the perspective of the Doppler velocity best estimate, as there are no independently measured sedimentation velocities available from surface-based observations. Therefore, it is assumed that good agreement in the Doppler velocities between EarthCARE CPR and surface-based radar implies a similarly reliable SVBE.

#### 2.3 Orbital-Radar simulator

To directly compare spaceborne measurements with surface-based observations, the Orbital-Radar tool (Pfitzenmaier et al., 2025), an instrument simulator specifically designed to convert high-resolution surface-based radar data into synthetic EarthCARE CPR primary measurements (e.g., reflectivity and Doppler velocity), is used. The tool integrates all key processing steps, including adjustments for differences in frequency bands, coordinate transformations, resolution-dependent integration and convolution, and realistic noise simulation. Specifically, the noise model estimates random errors associated with receiver noise, as well as the bias and uncertainty introduced by satellite motion. In addition, the tool explicitly diagnoses and flags NUBF, multiple scattering, and velocity folding, following the dedicated modules described in Pfitzenmaier et al. (2025).

When the input radar operates at a frequency different from CPR's W-band (94 GHz), for example, at Ka-band (35 GHz), reflectivity is converted to the 94 GHz scale using the formulation described by Kollias et al. (2019, Eq. (3)). This conversion is derived from the ice particle mass-size relation assumed in Mie scattering calculations. However, no correction for frequency differences is applied to Doppler velocities. For datasets that have not been pre-corrected for gaseous attenuation, a correction based on vertical water profiles is applied, although hydrometeor attenuation remains uncorrected due to the absence of detailed microphysical information (e.g., hydrometeor mass, density, and number concentration). This limitation may lead to discrepancies between reflectivities measured at different frequency bands, particularly in environments with strong liquid-induced attenuation.

The simulator converts surface-based radar time coordinates to along-track distance by multiplying with a constant horizontal wind speed (here, 9 m s<sup>-1</sup>). The data are then averaged along-track to match the CPR's horizontal resolution. This averaging accounts for both the sensor's instantaneous field of view (IFOV; approximately 750 – 800 m) and its 500 m integration interval. In the vertical dimension, convolution is performed over a 500 m pulse length with a 100 m sampling resolution using the CPR's distinctive asymmetric range weighting function that generates a sharp cutoff at the top of the pulse (Lamer et al., 2020). Notably, Doppler velocity convolution applies a reflectivity-weighted averaging approach, allowing stronger returns to have a greater influence.

Reflectivity uncertainty is estimated following the methods of Delanoë and Hogan (2010), where the uncertainty is derived using the SNR and the number of radar samples. For the Doppler velocity error, although the tool can reasonably estimate contributions from both satellite motion and receiver noise-related random error (i.e., thermal and speckle noise), our analysis incorporates only the satellite motion component to avoid overly noisy estimates. Given that our study focuses on comparing these simulations with the CPR L2 corrected Doppler velocity products (i.e., best estimates), using the noise-free Doppler variable is justified. Overall, the Orbital-Radar tool enables a direct and reliable comparison between spaceborne and surface-based radar measurements.

#### 2.4 Surface-based radar observations

Validation of the first spaceborne Doppler velocity measurements was performed by comparing them with observations from two high-latitude surface-based cloud profiling radars. The first dataset comes from the U.S. Department of Energy (DOE) Atmospheric Radiation Measurement (ARM) User Facility (Mather and Voyles, 2013; Kollias et al., 2020) at the North Slope of Alaska (NSA) site in Utqiagvik (71.34°N, 156.68°W), where a Ka-band (35 GHz) ARM Zenith Radar (KAZR) has been in continuous operation since May 2011. The second dataset is provided by the Aerosol, Clouds, and Trace Gases Research Infrastructure (ACTRIS) at the Neumayer site (70.67°S, 8.27°W) in Antarctica, managed by the German Alfred-Wegener Institute. This facility hosts a W-band (94 GHz) frequency-modulated continuous-wave (FMCW) cloud Doppler radar (Küchler et al., 2017) that has been operating since January 2024.

These two sites were selected for several key reasons. First, EarthCARE's sun-synchronous orbit results in a higher number of nearby overpasses; for example, the NSA site experiences roughly two to three times more visits than mid-latitude locations. Second, the two sites represent contrasting mispointing error regimes, enabling a comparative evaluation of the mispointing correction applied to the Doppler velocity measurement. A detailed description of this difference is provided in Section 3.1. Moreover, both locations frequently exhibit ice clouds at higher altitudes that are relatively free from liquid-induced attenuation, which provides favorable conditions for validation using two radars operating at different frequencies. Finally, low-level mixed-phase clouds are also common at high latitudes (Morrison et al., 2012). They create a particularly challenging environment for spaceborne Doppler velocity observations, which provides an opportunity for a more rigorous evaluation of the CPR's performance.

Surface-based radar observations from the NSA and Neumayer sites were processed using the Orbital-Radar tool to enable direct comparison with the CPR data. This study used data collected from June 2024 to July 2025, excluding the period from 23 June to 16 July 2024 due to the unavailability of EarthCARE CPR data. EarthCARE overpass events were defined as instances where the satellite passed within a 100 km radius of the surface-based radar site. In the coordinate transformation used by the Orbital-Radar tool, this 100 km along-track distance corresponds to approximately three hours, assuming a constant horizontal wind speed of 9 m s<sup>-1</sup>. Even if this coordinate transformation is not optimal for every overpass, with a sufficiently large sample the measurements are expected to converge to climatological values.

Although our focus is on Doppler velocities, differences in reflectivity between radar systems can introduce systematic biases in the reflectivity-weighted measurements. In this study, we use the corrected reflectivity from the CPR feature mask and reflectivity (C-FMR) product, based on the AC baseline. These reflectivities are feature-masked and corrected for gaseous attenuation. The sensitivity of this product is -35 dBZ in the troposphere. In contrast, surface-based radars tend to experience diminishing sensitivity with increasing range. For example, the NSA KAZR (in its general mode) exhibits lower sensitivity than CPR at altitudes above approximately 1 km (using only data with SNR > -15 dB). Meanwhile, at the Neumayer site, the FMCW radar has higher sensitivity up to around 10 km. To ensure a fair comparison and exclude low-quality data, each site retains only those measurements that exceed the higher of the two instruments' minimum detectable signal (MDS) thresholds. Additionally, to avoid contamination due to the CPR surface clutter, data below 600 m were also filtered out. We then calibrated the surface-based radar reflectivity using the CPR reflectivity as a reference, following the method of Kollias et al. (2019). In this calibration, only ice cloud above 3.5 km without a bright band were considered; profiles with any underlying cloud layers were excluded. As a result, calibration offsets of -2.1 dB for the NSA KAZR and -0.7 dB for the Neumayer FMCW radar were obtained, and these offsets were applied prior to processing with the Orbital-Radar tool.

An example of radar measurements of clouds observed by the CPR and the NSA site's surface-based radar (i.e., KAZR) during an EarthCARE overpass event on 7 October 2024 (UTC) is shown in Fig. 1. The radar observations reveal a two-layer cloud structure consisting of a low-level cloud with a cloud-top height around 1 km and a thick upper-level hydrometeor layer. The KAZR reflectivity (Fig. 1b) generally shows higher values compared to the CPR reflectivity (Fig. 1d) in areas of the upper-level hydrometeor layer exceeding 0 dBZ. This difference can largely be attributed to differences in radar frequencies; as ice crystals grow larger, the scattering regime shifts from Rayleigh ( $Z \propto D^6$ ) toward non-Rayleigh scattering where large ice particles contribute less ( $Z 

Figure 1: EarthCARE CPR observations with surface-based radar (i.e., KAZR) data from the NSA site during an EarthCARE overpass on 7 October 2024 (UTC). Panel (a) shows the satellite ground track over the NSA site, with a 100 km radius circle around the site. Panels (b)-(d) show radar reflectivity (Z), and panels (e)-(g) show Doppler velocity. Original measurements from the KAZR are presented in (b) and (e), with Doppler velocity sign inverted for consistency. Panels (c) and (f) show simulations from KAZR observations using the Orbital-Radar tool, with calibration applied to reflectivity. Panels (d) and (g) display EarthCARE CPR observations. A common minimum detectable signal (MDS) threshold is applied to both the simulated and EarthCARE CPR reflectivities; values below this threshold are shaded gray. Doppler velocities are shown only where reflectivity exceeds -15 dBZ. To avoid surface contamination, data below 600 m are excluded.

The original KAZR Doppler velocity measurements (Fig. 1e) are input to the Orbital-Radar tool to provide a Doppler velocity field equivalent to the CPR in terms of resolution, sensitivity, and uncertainty (Fig. 1f). The relatively small differences arise from three primary factors. First, the larger spatial resolution of CPR compared to KAZR removes variability at scales smaller than the CPR's sampling volume. Second, filtering procedures aimed at avoiding uncertainties in low SNR environments remove Doppler velocity data in regions where reflectivity falls below -15 dBZ. This filtering particularly leads to information loss at cloud boundaries and in regions dominated by smaller ice particles. Finally, additional data loss occurs below 600 m altitude due to surface clutter filtering.

After these processing steps, the surface-based radar depiction of the ice cloud layer (Fig. 1c and 1f) can be compared to the direct measurements from the CPR (Fig. 1d and 1g). The two radars detect similar vertical extents of the upper-layer cloud, showing good agreement on cloud top (8.5 km) and bottom (3 km) heights after applying the common MDS (see Fig. 1c and 1d). The Doppler velocity structures exhibit a generally consistent pattern, with both radars predominantly observing downward mean Doppler velocities with magnitudes greater than 0.8 m s<sup>-1</sup> within the altitude layer between 3.5 km to 5.5 km and lower velocities appear elsewhere (see Fig. 1f and 1g).

However, near the CPR's Doppler velocity measurement limit (i.e., reflectivity of -15 dBZ), uncertainties increase, resulting in locally enhanced variability in CPR Doppler velocity data (e.g., at cloud boundaries shown in Fig. 1g). According to the Japan Aerospace Exploration Agency's (JAXA's) release notes for the EarthCARE CPR Level-1 product (JAXA, 2025), this issue arises due to a technical problem related to an imbalance in the CPR's in-phase/quadrature (I/Q) channels, affecting measurements collected prior to 5 December 2024. Consequently, Doppler velocity data with reflectivity below -10 dBZ from before this date should strictly be considered unreliable. Nevertheless, to avoid substantial sample loss, this study adopts a lower reflectivity threshold of -15 dBZ and carefully considers the associated uncertainties during analysis. After 5 December 2024, this technical issue was resolved, significantly improving the CPR's Doppler velocity measurement limit down to reflectivities as low as -21 dBZ. Despite this improvement, since a considerable portion of data analyzed in this study was collected before this fix, the threshold of -15 dBZ is consistently maintained for the entire dataset.

#### 3 Results and discussion

215

220

225

230

235

In this section, the paired datasets collocated from the two high-latitude locations are classified into two distinct cloud regimes. The first category focuses on mid- and upper-level ice cloud layers located above 3.5 km altitude. Liquid water in these layers is infrequently observed (Kollias et al., 2019). To minimize the influence of liquid contamination, we further excluded all cases in which a bright band was detected, as this indicates the presence of liquid water. We also excluded profiles where underlying cloud layers were present beneath the ice clouds. As a result, the selected layers contain limited or no liquid clouds, minimizing attenuation effects. The validation of the CPR Doppler velocities for these ice cloud cases is discussed in Section 3.1.

The second category consists of low-level clouds whose reflectivity-defined cloud top remains below 2.5 km. At high latitudes, such clouds are often mixed-phase clouds containing supercooled liquid, and they are particularly frequent in the Arctic (Zhang et al., 2019). Section 3.2 addresses the challenges and limitations of observing Doppler velocities from space within low-level mixed-phase cloud environments, while also discussing its potential to provide new insights under conditions where the CPR performs reliably.

#### 3.1 Validation of CPR Doppler velocity in ice clouds

Ice clouds are the most commonly observed cloud type at higher altitudes in high-latitude regions, primarily due to the cold atmospheric conditions (Shupe, 2011; based on a few Arctic ground-based sites). Stratiform ice clouds typically exhibit similar

fall-speed distributions at both Arctic and Antarctic sites. Smaller ice crystals, predominantly near cloud tops, fall slowly—typically at around  $0.1 - 0.5 \text{ m s}^{-1}$ —while larger particles near the cloud base often fall faster, exceeding  $1 \text{ m s}^{-1}$  and occasionally reaching up to  $2 \text{ m s}^{-1}$ . Velocities above  $2 - 3 \text{ m s}^{-1}$  are rare unless liquid-phase or precipitation processes dominate. On average, sedimentation velocities of ice clouds range between  $0.5 - 1.0 \text{ m s}^{-1}$  (Heymsfield and Westbrook, 2010; Protat and Williams, 2011; Tridon et al., 2022; Wiener et al., 2024).

Figure 2: Validation of EarthCARE CPR Doppler velocities at the NSA (top panels) and Neumayer (middle panels) sites. Panels (a)-(c) and (e)-(g) show Doppler velocity CFADs with median profiles (solid lines) and 25th and 75th percentiles (dotted lines), while (d) and (h) show mean profiles (thick colored lines; see legend) along with  $\pm 1$  standard deviation (horizontal bars). Gray shaded areas indicate height levels with sufficient data ( $\geq 3\%$  of total samples). Surface-based radar Doppler velocities, converted to CPR-equivalent values, are shown in panels (a) and (e). Panels (b) and (f) present CPR data before pointing bias ( $e_p$ ) correction, and (c) and (g) after the correction. All panels showing vertical distributions use a height bin size of 0.5 km. Panel (i) shows the time series of the pointing bias, with circles representing the NSA site and triangles representing the Neumayer site. Black-filled symbols correspond to ascending-orbit passes, while gray-filled symbols to descending-orbit passes.

Figure 2 (top and middle rows) shows contoured frequency by altitude diagrams (CFADs) of Doppler velocity and their corresponding mean profiles for ice clouds above 3.5 km altitude at the NSA (Arctic) and Neumayer (Antarctica) sites. Only Doppler velocity measurements associated with reflectivity values greater than -15 dBZ are included. At both sites, surface-based radar observations (Fig. 2a, e) show that the median Doppler velocity gradually decreases from about 0.7 m s<sup>-1</sup> near 3.5 km altitude to about 0.5 m s<sup>-1</sup> around 7 km. Considering typically weak vertical air motions in these clouds, these values align well with known sedimentation velocities.

However, Doppler velocities measured by CPR prior to correcting for pointing bias  $(e_p)$  (Fig. 2b, f) differ significantly from those measured at the surface. Notably, the impact of the mispointing appears differently at the two sites. At NSA, the mean Doppler velocity difference between CPR and surface radar is about  $0.4 \text{ m s}^{-1}$ , whereas at Neumayer, the mean difference is almost negligible, particularly below 6 km altitude, albeit with a broader spread in CPR measurements. These discrepancies stem from variations in  $e_p$  illustrated in Fig. 2i.

265

270

Rapid temperature fluctuations caused by sunlight exposure induce thermoelastic distortions in the CPR antenna (see Puigdomènech et al., 2025), making  $e_p$  seasonal- and latitude-dependent. Thus, distinct variations emerge also between ascending and descending passes. Although the time differences between these passes is typically only 9–12 hours at around 70° latitude, the associated solar illumination conditions can be nearly opposite, sometimes resulting in greater differences in  $e_p$  than those seen across monthly timescales. At the NSA site,  $e_p$  remains consistently negative throughout the observation period, peaking around  $-0.7 \text{ m s}^{-1}$  (ascending) in October 2024 and reducing to about  $-0.1 \text{ m s}^{-1}$  (descending) in June and July 2025. In contrast,  $e_p$  at Neumayer transitions from negative to positive around mid-September 2024 for descending passes and mid-November 2024 for ascending passes (Fig. 2i). They then reach their positive maxima in March 2025 (descending) and January 2025 (ascending), before decreasing again, with the ascending passes even returning to negative values. These seasonal- and latitude-dependent variations in  $e_p$  explain the differences between CPR Doppler velocities before and after the mispointing correction (red and orange lines in Fig. 2d and Fig. 2h).

After applying pointing bias corrections (Fig. 2c, g), CPR Doppler velocities closely align with surface radar observations (Fig. 2a, e). The corrected CPR mean profiles (red lines in Fig. 2d, h) exhibit strong agreement with those from surface-based radars (blue lines), particularly within the altitude range shaded gray, where each 500 m bin contains at least 3% of the total valid Doppler velocity samples in both datasets. Outside this shaded region, limited sample sizes tend to amplify discrepancies between the two radars, so these areas were excluded from the validation analysis. Quantitatively, the mispointing correction substantially improves the CPR Doppler velocity accuracy. At NSA, the mean velocity bias is reduced from 0.388 m s<sup>-1</sup> before correction to less than 0.02 m s<sup>-1</sup> afterward. Similarly, at Neumayer, the bias drops from 0.02 m s<sup>-1</sup> to less than 0.01 m s<sup>-1</sup> following correction. At both sites, moreover, the standard deviations within the gray-shaded range also closely match, indicating that CPR captures the Doppler variability well.

The reflectivity-velocity (Z-V) relationships derived from CPR and surface-based radar measurements at each site are shown in Fig. 3. At both sites, the median Doppler velocity from the surface-based radars gradually increases with reflectivity, ranging

from near  $0.5 \text{ m s}^{-1}$  at -15 dBZ to about  $0.7 - 0.8 \text{ m s}^{-1}$  at -3 dBZ (see Fig. 3). While velocities at Neumayer tend to be marginally lower compared to NSA, the difference remains within  $0.05 \text{ m s}^{-1}$  for reflectivities between -15 and -3 dBZ. Outside the region of sufficient sampling, defined by gray shading in Fig. 3, statistical reliability decreases due to limited observations, leading to discrepancies exceeding  $0.1 \text{ m s}^{-1}$  between the two sites. Therefore, Doppler velocity validation is conducted only within this reliably sampled reflectivity range.

Figure 3: Distribution of Doppler velocities as a function of reflectivity at the NSA and Neumayer sites, shown as box-and-whisker plots. Boxes indicate median values and lower and upper quartiles; whiskers represent minimum and maximum values excluding outliers, calculated within reflectivity bins of 1.0 dB. Probability density functions (PDFs) for reflectivity (top sub-panels) and Doppler velocity (right sub-panels) are provided for reference. Three datasets are compared: blue represents surface-based radar measurements converted to CPR-equivalent values, yellow shows CPR measurements that include pointing bias, and red corresponds to the data after correction. Gray shaded areas highlight reflectivity bins with sufficient data density ( $\geq 3\%$  of total samples).

Within this reflectivity range, CPR Doppler velocities prior to correcting for pointing bias are systematically underestimated compared to surface-based radar observations at NSA (left panel of Fig. 3), whereas at Neumayer (right panel of Fig. 3), the CPR Doppler velocity distribution shows a notably broader spread than the surface-based radar. After applying the mispointing correction, CPR velocity biases and uncertainties are significantly reduced at both sites, with the mean difference of median Doppler velocities decreasing from 0.376 m s<sup>-1</sup> to less than 0.01 m s<sup>-1</sup> at NSA and from 0.014 m s<sup>-1</sup> to less than 0.01 m s<sup>-1</sup> at Neumayer. The results additionally confirm the absence of significant bias even at low SNR conditions. Together with the validation results using vertical mean profiles presented in Fig. 2, these outcomes demonstrate that the mispointing correction enhances the accuracy of CPR Doppler velocities, confirming reliability at the level of a few centimeters per second for climatological estimates of sedimentation velocities.

Overall, the comparison demonstrates that the quality-controlled Doppler velocities from the CPR L2 product exhibit minimal biases (on the order of centimeters per second), confirming that the implemented mispointing correction effectively ensures reliable Doppler velocity measurements. However, it should be noted that the validation presented here is limited to stratiform

ice clouds with gentle vertical and horizontal gradients of radar reflectivity and Doppler velocity. Different cloud systems may introduce other significant error sources. For example, aliasing and multiple scattering may become important in strong convective systems (Galfione et al., 2025). More importantly, in cloud and precipitation layers where rapid ice particle growth in size and/or density occurs, the 500 m pulse length of the CPR is expected to smooth the radar signature of these microphysical changes in the ice particles properties. Low-level mixed-phase clouds are a great example to illustrate these effects.

## 3.2 CPR Doppler velocity in low-level mixed-phase clouds

Unlike Section 3.1, which focused on validating CPR Doppler velocities in ice clouds, the goal here is to assess the CPR's ability and limitations in observing Doppler velocities in low-level mixed-phase clouds. These clouds present specific challenges for CPR Doppler velocity measurements. In addition to the well-known issue of surface clutter, which typically affects altitudes up to 500 – 600 m over the ocean, these clouds themselves have structural features complicating their accurate observation. Field campaign results suggest that these clouds typically consist of both supercooled liquid droplets and ice particles within the cloud layer, overlying precipitating ice particles (Tan et al., 2023). These clouds are generally shallow in vertical extent (~500 m), occasionally extending up to 3 km (Shupe et al., 2008; de Boer et al., 2009; Zhang et al., 2019). Despite their limited thickness, these clouds frequently display strong vertical gradients in microphysical properties. For example, ice particle fall velocities are typically low (often less than 0.5 m s<sup>-1</sup>) near the cloud top but rapidly increase toward the cloud base, reaching approximately 1 – 2 m s<sup>-1</sup> or even higher (Shupe et al., 2008). Such increases in fall velocity reflect particle growth processes, including aggregation and riming (Chellini and Kneifel, 2024). In particular, riming tends to make particles heavier and denser more quickly, leading to a more rapid increase in fall velocity.

The CPR's vertical pulse length of 500 m is comparable to the typical thickness of these mixed-phase clouds. Consequently, the Doppler velocities measured by the CPR inherently represent vertically averaged signals over this vertical range, thereby smoothing out sharp gradients and potentially obscuring detailed cloud structures. The Orbital-Radar tool introduces the CPR pulse length through vertical convolution of the original resolution KAZR radar data with the CPR Point Target Response (PTR, Lamer et al., 2020; Pfitzenmaier et al., 2025).

Figure 4 illustrates Doppler velocity distributions measured by CPR and KAZR at the NSA site for low-level clouds with cloud tops defined below 2.5 km. Differences in Doppler velocities measured by CPR and KAZR can largely be categorized into two primary factors. First, the different operating frequencies of CPR (94 GHz) and KAZR (35 GHz) can induce a differential Doppler velocity (DDV), reflecting how ice particles backscatter differently at these two bands (Matrosov, 2017). Although this difference is often negligible in ice clouds (Courtier et al., 2024), the riming process in mixed-phase clouds can increase particle size, and DDV values can reach up to 0.8 m s<sup>-1</sup> depending on the degree of riming (Oue et al., 2021; Kollias et al., 2022). Note that the Orbital-Radar tool does not correct for frequency differences in Doppler velocity, so DDV can be observed.

Figure 4: Box-and-whisker plots of Doppler velocity distributions for low-level clouds observed at the NSA site. Cloud profiles were selected based on a reflectivity-defined cloud top below 2.5 km; for each profile, the highest valid Doppler velocity measurement is set as the reference level (0 m), and successive 100 m layers down to -500 m were plotted. Light blue and deep blue box plots represent KAZR (35 GHz) Doppler velocities before and after reflectivity-weighted range convolution, respectively, while red box plots represent EarthCARE CPR (94 GHz) Doppler velocity measurements.

Another factor is that the CPR pulse length significantly impacts Doppler velocity measurements near cloud tops. KAZR Doppler velocities, processed to the CPR's vertical sampling resolution of 100 m (light blue in Fig. 4), exhibit median values of 0.25 m s<sup>-1</sup> at cloud tops but increase rapidly to 0.86 m s<sup>-1</sup> within just 500 m below. However, when these velocities undergo reflectivity-weighted range convolution to match the CPR's vertical resolution of 500 m, the lower velocities near cloud tops are combined with higher velocities from lower altitudes. Consequently, the median Doppler velocity at cloud top increases from 0.25 m s<sup>-1</sup> to 0.44 m s<sup>-1</sup> after convolution (deep blue in Fig. 4). Thus, the CPR's measured Doppler velocities (red in Fig. 4) at cloud top, with a median of 0.45 m s<sup>-1</sup>, are likely inflated due to vertical smoothing, and the magnitude of this effect depends on the strength of vertical velocity gradients within the clouds. However, the distortion caused by the CPR pulse length effect becomes negligible starting from about 300 m below the cloud top, where vertical gradients are weaker. Figure 5 presents mean reflectivity and Doppler velocity profiles measured by CPR and surface-based radars at both sites for these low-level clouds. Due to the limited vertical extent of low-level clouds, a vertical bin size of 100 m was used for the analysis. Additionally, the mean liquid water path (LWP) for low-level cloud profiles was computed using microwave radiometer 3-channel (MWR3C) data at NSA and the humidity and temperature profiler (HATPRO) data at Neumayer, with cases categorized into low LWP (≤ 100 g m<sup>-2</sup>) and high LWP (> 100 g m<sup>-2</sup>) conditions. Since EarthCARE does not provide LWP observations, the classification here is based on case-mean LWP derived from the surface-based measurements. As a result, the "low LWP" and "high LWP" groups should be regarded as rough division.

Figure 5: Mean reflectivity (top row) and Doppler velocity (bottom row) profiles for low-level clouds at NSA and Neumayer, with  $\pm 1$  standard deviation (horizontal bars). Cases are grouped by mean liquid water path (LWP):  $\leq 100$  g m<sup>-2</sup> (first and third columns) and > 100 g m<sup>-2</sup> (second and fourth columns). Note that the surface-based radars operate at different frequencies: the KAZR at 35 GHz for NSA, and the FMCW radar at 94 GHz for Neumayer. Only measurements above 600 m altitude are included. Gray shading marks height levels with sufficient data ( $\geq 3\%$  of total samples).

At the Neumayer site, where the surface radar operates at the same frequency as CPR (94 GHz), the mean Doppler velocity differences between the two radars remain below 0.05 m s<sup>-1</sup> within the gray-shaded altitude range, regardless of LWP conditions (Fig. 5g and 5h). This consistency indicates that CPR remains highly reliable in observing Doppler velocities in low-level clouds as well. However, at the NSA site, where surface radar operates at a lower frequency (35 GHz), Doppler velocity differences depend on the degree of riming. Under low LWP conditions (Fig. 5e), the difference between CPR and KAZR remains sufficiently small to be considered negligible below 1.5 km altitude. Conversely, under high LWP conditions (Fig. 5f), the Doppler velocity differences increase up to about 0.3 m s<sup>-1</sup>. The magnitude of these differences aligns with moderate riming scenarios reported in previous studies (Oue et al., 2021; Kollias et al., 2022), although it can vary depending on particle size, density, and shape. Additionally, stronger liquid attenuation at 94 GHz (see Fig. 5b) causes CPR to miss weak reflectivities that KAZR still detects, and this sampling difference may have slightly contributed to the mean Doppler velocity differences between the two radars (Fig. 5f).

In summary, CPR Doppler velocity measurements in low-level mixed-phase clouds have three limitations: (1) contamination by surface clutter at altitudes between 500 - 600 m above the surface, (2) large uncertainties near cloud tops due to low SNR conditions, and (3) distortion caused by the PTR effect (i.e., vertical smoothing) at cloud tops under strong vertical gradients. Therefore, Doppler velocities within these layers require careful interpretation or exclusion from analysis. Nonetheless, outside these problematic layers, CPR Doppler velocities exhibit high reliability even at a vertical sampling resolution of 100 m.

Furthermore, the combination of CPR data with surface-based radars operating at different frequencies provides additional microphysical information related to particle growth processes like riming.

## 395 4 Conclusions

This study presents the first validation results of the EarthCARE CPR L2 Doppler velocity product, referred to as the "Doppler velocity best estimate," using surface-based radar observations from two high-latitude sites: NSA (71.34°N, 156.68°W) and Neumayer (70.67°S, 8.27°W). The surface-based observations were processed into synthetic CPR products using the instrument simulator, Orbital-Radar tool (Pfitzenmaier et al., 2025), which accounts for coordinate transformation, frequency conversion, resolution matching, and the simulation of relevant noise. Between June 2024 and July 2025, only ice cloud cases located above 3.5 km (unaffected by liquid-induced attenuation) were selected for validation, based on observations obtained within a 100 km radius of each site. Prior to validation, reflectivity calibration was applied based on CPR measurements, with a calibration offset of -2.1 dB for the KAZR at NSA and -0.7 dB for the FMCW at Neumayer. Subsequently, the same MDS was applied to the reflectivity data from both the CPR and the surface-based radars.

- The validation results indicate that the CPR L2 Doppler velocities exhibit near-zero biases. These results demonstrate that the implemented pre- and post-processing corrections (Kollias et al., 2023; Puigdomènech et al., 2025) effectively reduce Doppler velocity biases. In particular, this study confirms that the antenna mispointing bias, arising from the satellite's high orbital speed and antenna thermoelastic distortions due to different sunlight illumination, was successfully corrected. These findings provide strong support for the global application of the first spaceborne Doppler velocity products.
- In addition, we evaluated the performance of the CPR Doppler velocity in low-level mixed-phase clouds. These clouds pose a significant challenge for CPR Doppler velocity measurements due to their strong vertical gradients over a shallow vertical extent (typically around 500 m). Near the cloud tops, where supercooled liquid and small ice particles are present, Doppler velocity measurements are often limited due to low SNR, and even when detection is possible, the measurements tend to be inflated due to PTR effects amplified by the strong vertical gradients. Despite these limitations, CPR Doppler velocities remain reliable in low-level mixed-phase clouds a few layers below the cloud top, excluding layers affected by surface clutter. Furthermore, this study demonstrated that when CPR (94 GHz) is used in conjunction with surface-based radars operating at different frequencies (e.g., 35 GHz), DDV signals are observed due to differences between the Rayleigh and non-Rayleigh scattering regimes. These DDV signals, when used alongside Doppler velocities obtained from single-frequency radars, may provide a complementary means for estimating climatologies of the degree of riming in ice particles.
- Overall, this study demonstrates that, with appropriate corrections, EarthCARE CPR can reliably measure Doppler velocities at least in ice clouds, and even in challenging environments such as low-level mixed-phase clouds, provided that certain limitations identified in this study are taken into account. Therefore, this study not only supports the reliability of the year-round, global hydrometeor sedimentation velocity data to be provided for the first time by EarthCARE, but also reinforces its

credibility as an observational constraint when applied to cloud microphysical parameterizations in numerical weather prediction and climate models.

## **Author contributions**

JK performed the analysis, curated the data, created the figures, and wrote the draft of the manuscript. PK provided supervision and reviewed the manuscript. BPT provided the EarthCARE data and reviewed the manuscript. AB and IT provided feedback on the results and the manuscript.

## 430 Competing interests

None of the authors has any competing interests.

## Acknowledgements

Work done by JK, PK, and BPT were supported by the European Space Agency (ESA) under the Clouds, Aerosol, Radiation

– Development of INtegrated ALgorithms (CARDINAL) project (RFQ/3-17010/20/NL/AD). JK was also supported by

Natural Sciences and Engineering Research Council of Canada (NSERC) Discovery Grant RGPIN-2021-02720. PK and BPT

were supported by the National Aeronautics and Space Administration (NASA) under the Atmospheric Observing System

(AOS) project (Contract number: 80NSSC23M0113).

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
