# Peer review of "Evaluation of the EarthCARE Cloud Profiling Radar (CPR) Doppler velocity measurements using surface-based observations"

_EGUsphere, 2025_

## Referee Comment (RC1)

**Review of Evaluation of the EarthCARE Cloud Profiling Radar (CPR) Doppler velocity measurements using surface-based observations, by Kim et al.**

This paper presents the first ever evaluation of the first ever 94 GHz Doppler measurements in space. In that regard, this paper will become a very important reference paper once published for a very large community of EarthCARE users. The paper is extremely well written, and the evaluation presented is thorough and convincing. I only have comments aimed at improving the description of the evaluation results, and I suggest the paper should be published once these relatively minor comments are addressed.

**Comments:**

1. Section 2.1: since you have to do do the same thing for both the ground based and spaceborne measurements, I don't really see the point of describing the relationship between terminal fall speed and Doppler velocity. In ground – satellite comparisons, you would find that the errors on Doppler are the same as the errors on fall speed, pretty much (that is also what is discussed in lines 110-113 by the authors themselves), and as a matter of fact, you don't compare fall speeds in the results section. Where is the added value of that section?

2. Line 124: I think it would be nice to describe in one sentence how this Kollias (2019) correction works. Also, I assume you are not correcting Doppler for the frequency difference? Maybe it would be worth adding that because as I was reading the results section, I wondered.

3. Line 130 (and line 165): what is the justification for 9 ms-1 ? Can't you use vertical profiles of horizontal winds that are changing from one day to another to be more accurate with this time – space conversion (wind profiler, Doppler lidar, radiosonde interpolations at the ARM site, even NWP, etc …). I wonder how much difference it would make.

4. Lines 155 – 159: Isn't there a contradiction here? Polar environments are indeed known to host the highest frequency of occurrence of mixed-phase and supercooled clouds that attenuate the W-band signal, but you start this paragraph by saying the opposite.

5. Line 163: So you use a 00km radius around the site and you develop some statistical corrections, which is fine. I have two comments / suggestions about that. Do you also have very close overpasses? Would it be useful to do a separate analysis of these near perfect collocations? Also, I was wondering if you could show comparisons of CFADs of reflectivity (after using the radar simulator) to make sure you are working with similar enough cloud statistics?

6. Figure 1: How close to the site is the closest profile of that figure and where is it on the cross-section ? I think there would be value in adding a panel where you compare the closest profiles of reflectivity and Doppler velocity together too (if the satellite was very close).

7. Lines 229-230: Are you stating that by looking at lidar data within these clouds ? The problem is that the lidar measurements are often extinguished before reaching these clouds by low-level liquid layers. My experience from the Southern Ocean tells me that this statement is not correct and that the impact of liquid attenuation cannot be discarded so easily in such comparisons.

8. Line 231: Removing clouds using bright band detection is a good idea, but when using this you are excluding precipitating cases with liquid and ice phases (frontal systems), not SLW or mixed-phase clouds.

9. Analysis of Figure 2 results: I think another important (and remarkable) point to make from this figure is that the standard deviation is very similar at all heights where you have enough samples. Doppler velocities from satellite are not only virtually unbiased estimates but do capture the Doppler variability accurately. I would highlight that too.

10. Line 346: So just to be sure, you are not trying to correct for frequency-induced Doppler differences in the radar simulator? This should be clearly explained both in the description of the simulator and maybe here as well.

11. Lines 370-371: that's really a missed opportunity not to compare cloud results for LWP >100 from that site. That was the perfect missing piece of information to infer if the differences you see at NSA are really due to the frequency difference or any other issue! I'd recommend doing that for sure or explore other sites where you have more data like that at 94 GHz. The reader here is left wondering if there's a problem or if that's just a limitation of these comparisons, but then if it is just a limitation, it's not a very interesting evaluation.

12. Figure 5: do you still use the 3% of total number of data points to define what is statistically significant or not ?

13. Line 385: Related to my point 11 above, the problem with this comment is that you have the difference in Doppler due to different frequencies that could mask some other issues. This is why it would seem quite important to do the comparisons at 94 GHz with a ground-based radar for LWP > 100. Also it would seem important to compare reflectivities too to document any statistical effect from liquid attenuation in these comparisons.

*Good luck with the revision,*

*Alain Protat*

*Bureau of Meteorology, Melbourne, Australia*

*20/07/2025.*

---

## Author Comment (AC1)

**Final author response to referee (egusphere-2025-2697)**

RC1: "Comment on egusphere-2025-2697", Alain Protat (Referee #1)

**General comments**

This paper presents the first ever evaluation of the first ever 94 GHz Doppler measurements in space. In that regard, this paper will become a very important reference paper once published for a very large community of EarthCARE users. The paper is extremely well written, and the evaluation presented is thorough and convincing. I only have comments aimed at improving the description of the evaluation results, and I suggest the paper should be published once these relatively minor comments are addressed.

The authors sincerely thank the reviewer for the time and effort put into reviewing our manuscript and for the constructive comments. We also appreciate the recognition of the value of our study. The reviewer's feedback has helped us to improve the quality and clarity of our work.

The authors have carefully considered all the suggestions. Before providing detailed point-by-point responses to each comment, we would like to highlight one major change implemented in the revised manuscript. The observation period used in this study has been extended from June 2024 – February 2025 (~8 months) to June 2024 – July 2025 (~13 months). This update was made to address the reviewer's concerns raised in Comments #11 and #13, which required the use of additional dataset. We considered it more appropriate to update the entire dataset to maintain consistency across the manuscript. As a result, all figures have been updated, and the related text has been revised as well. Further details can be found in the revised manuscript.

**Specific comments**

**Comment #1**

Section 2.1: since you have to do the same thing for both the ground based and spaceborne measurements, I don't really see the point of describing the relationship between terminal fall speed and Doppler velocity. In ground – satellite comparisons, you would find that the errors on Doppler are the same as the errors on fall speed, pretty much (that is also what is discussed in lines 110-113 by the authors themselves), and as a matter of fact, you don't compare fall speeds in the results section. Where is the added value of that section?

The reason we included a description of the relationship between hydrometeor sedimentation velocity and Doppler velocity in this manuscript is that Doppler measurements, which have so far been mainly

used within the radar community, will be utilized by a much broader research community through EarthCARE. The mission provides the first-ever global observations of sedimentation velocity, enabling the evaluation of fall speed parameterizations in weather and climate models, which is one of the mission's central objectives. In this context, we considered it important to link the Doppler velocities validated in this study to the sedimentation velocities that a wider community will be interested in.

**Comment #2**

Line 124: I think it would be nice to describe in one sentence how this Kollias (2019) correction works. Also, I assume you are not correcting Doppler for the frequency difference? Maybe it would be worth adding that because as I was reading the results section, I wondered.

We have revised Section 2.3 to add a one sentence description of the correction work and state that Doppler velocities are not corrected for frequency differences. The revised text reads:

(lines 119 - 122) "When the input radar operates at a frequency different from CPR's W-band (94 GHz), for example at Ka-band (35 GHz), reflectivity is converted to the 94 GHz scale using the formulation described by Kollias et al. (2019, Eq. (3)). This conversion is derived from the ice particle mass-size relation assumed in Mie scattering calculations. However, no correction for frequency differences is applied to Doppler velocities."

**Comment #3**

Line 130 (and line 165): what is the justification for 9 ms-1 ? Can't you use vertical profiles of horizontal winds that are changing from one day to another to be more accurate with this time – space conversion (wind profiler, Doppler lidar, radiosonde interpolations at the ARM site, even NWP, etc …). I wonder how much difference it would make.

The Orbital-Radar tool used in this study assumes a constant horizontal wind speed (6 m s-1 by default) throughout the whole atmosphere to perform the time – space conversion. To better represent high-latitude conditions, we updated the default to 9 m s-1 using radiosonde winds averaged below 10 km. Under this setting, 100 km corresponds to about 3 hours.

We agree that using winds that vary with height and time could refine individual collocations. However, vertical wind shear makes the conversion challenging, and the satellite ground track is generally not aligned with the cloud advection vector, meaning that even with a sophisticated conversion, the comparison of the same cloud cannot be guaranteed. Given our objective of climatological validation, the added complexity does not provide substantial benefits.

Our assumption was that, given a sufficiently large sample, the mean vertical profiles of Z and V, as well as the Z-V relationship, would converge robustly, with random variability canceling out. This assumption was indeed supported by our results. When we extended the dataset from about 8 months to 13 months during the revision process, the mean Z and V profiles and Z-V relationships showed little change.

We added the following sentence in the revised manuscript.

(lines 164-165) "Even if this coordinate transformation is not optimal for every overpass, with a sufficiently large sample the measurements are expected to converge to climatological values."

**Comment #4**

Lines 155 – 159: Isn't there a contradiction here? Polar environments are indeed known to host the highest frequency of occurrence of mixed-phase and supercooled clouds that attenuate the W-band signal, but you start this paragraph by saying the opposite.

In this context, "ice clouds" refers to ice clouds at higher altitudes, such as cirrostratus, altostratus, and anvil outflows. Although low-level mixed-phase clouds are common in polar environments, ice clouds occurring at mid- and upper-levels are observed more frequently.

We have undated the sentence in the revised manuscript:

(lines 154-156) "Moreover, both locations frequently exhibit ice clouds at higher altitudes that are relatively free from liquid-induced attenuation, which provides favorable conditions for validation using two radars operating at different frequencies."

**Comment #5**

Line 163: So you use a 100km radius around the site and you develop some statistical corrections, which is fine. I have two comments / suggestions about that. Do you also have very close overpasses? Would it be useful to do a separate analysis of these near perfect collocations? Also, I was wondering if you could show comparisons of CFADs of reflectivity (after using the radar simulator) to make sure you are working with similar enough cloud statistics?

First, over about 13 months at the NSA site, we identified 184 overpasses within 100 km of the site (including nearly cloud-free cases). Of these, 16 had the closest approach of 5 km or less, and only two contained a detectable cloud layer several kilometers deep at the overpass point. This sample is too small for a separate, statistically robust analysis. More importantly, the near-zero bias we find in CPR Doppler velocities is meaningful at a statistical level. With very small samples, it is difficult distinguish between uncertainty and bias

Second, we have added CFAD comparisons of reflectivity in Figure R1 (below).

[Figure]

Figure R1: CFADs of radar reflectivity at NSA (Arctic; top) and Neumayer (Antarctica; bottom). Left panels show surface-based radar reflectivities processed with the Orbital-Radar tool and offset-calibrated against CPR reflectivity. Right panels show EarthCARE CPR reflectivity. The thick black dashed curve denotes the minimum detectable signal (MDS).

**Comment #6**

Figure 1: How close to the site is the closest profile of that figure and where is it on the cross-section ? I think there would be value in adding a panel where you compare the closest profiles of reflectivity and Doppler velocity together too (if the satellite was very close).

The closest satellite profile in Figure 1 is 2.7 km from the site. On the cross section, the overpass point corresponds to 0 km on the relative along-track axis. Although this distance is relatively close, we decided not to include an additional panel showing the closest reflectivity and Doppler velocity profiles. This study does not recommend direct comparisons of single instantaneous Doppler velocity profiles. The accuracy of spaceborne Doppler velocity is affected by random errors induced by Doppler broadening, which can remain on the order of ~0.5 m s-1 even after applying corrections, as described in the manuscript. Our objective is to validate performance at a statistical level, and the results confirm that despite such uncertainty, there is no systematic bias in CPR Doppler velocities.

**Comment #7**

Lines 229-230: Are you stating that by looking at lidar data within these clouds? The problem is that the lidar measurements are often extinguished before reaching these clouds by low-level liquid layers. My experience from the Southern Ocean tells me that this statement is not correct and that the impact of liquid attenuation cannot be discarded so easily in such comparisons.

We did not use lidar data in this study. In our original manuscript, lines 229-230 states that "The first category focuses on mid- and upper-level ice cloud layers located above 3.5 km altitude. Supercooled liquid water in these layers in infrequently observed." Here, the term "Liquid water" should be used

instead of "Supercooled liquid water." This statement is justified by the fact that, at these latitudes, the melting layer is generally located below 3.5 km (Kollias et al., 2019).

The reviewer's concern seems to be that lower-level liquid layers and separate low-level mixed-phase clouds may cause attenuation in surface radar observations. In this study, cases with a detected melting layer were excluded, which means that clouds including lower-level liquid layers were not included. However, we agree that low-level mixed-phase clouds situated below ice clouds can attenuate the surface-based radar reflectivity.

Therefore, when we expanded the dataset to cover the 13-month period, we additionally excluded such profiles. We also added the following sentence in the revised manuscript:

(lines 225-226) "We also excluded profiles where underlying cloud layers were present beneath the ice clouds."

**Comment #8**

Line 231: Removing clouds using bright band detection is a good idea, but when using this you are excluding precipitating cases with liquid and ice phases (frontal systems), not SLW or mixed-phase clouds.

We agree with the reviewer's observation, and we believe that our response to Comment #7 addresses this concern.

**Comment #9**

Analysis of Figure 2 results: I think another important (and remarkable) point to make from this figure is that the standard deviation is very similar at all heights where you have enough samples. Doppler velocities from satellite are not only virtually unbiased estimates but do capture the Doppler variability accurately. I would highlight that too.

We have added the following sentence in the revised manuscript.

(lines 282-283) "At both sites, moreover, the standard deviations within the gray-shaded range also closely match, indicating that CPR captures the Doppler variability well."

**Comment #10**

Line 346: So just to be sure, you are not trying to correct for frequency-induced Doppler differences in the radar simulator? This should be clearly explained both in the description of the simulator and maybe here as well.

We have added the following sentence to the revised manuscript.
(lines 343-344) "Note that the Orbital-Radar tool does not correct for frequency differences in Doppler velocity, so DDV can be observed."

**Comment #11**

Lines 370-371: that's really a missed opportunity not to compare cloud results for LWP >100 from that site. That was the perfect missing piece of information to infer if the differences you see at NSA are really due to the frequency difference or any other issue! I'd recommend doing that for sure or explore other sites where you have more data like that at 94 GHz. The reader here is left wondering if there's a problem or if that's just a limitation of these comparisons, but then if it is just a limitation, it's not a very interesting evaluation.

We agree with the reviewer's concern. To address this point, we extended the observation period in this study from June 2024 - February 2025 (~8 months) to June 2024 - July 2025 (~13 months). This allowed us to include additional cases under high LWP conditions at Antarctica, providing the most straightforward way to validate Doppler velocities at the same frequency as CPR. As expected, no significant bias was found. Please refer to the updated Figure 5 in the revised manuscript. These results support our argument that the differences observed at NSA under high LWP conditions are primarily due to the frequency difference.

**Comment #12**

Figure 5: do you still use the 3% of total number of data points to define what is statistically significant or not?

Yes, we still use the 3% threshold to define sufficient sampling. We have clarified this in the Figure 5 caption.

**Comment #13**

Line 385: Related to my point 11 above, the problem with this comment is that you have the difference in Doppler due to different frequencies that could mask some other issues. This is why it would seem quite important to do the comparisons at 94 GHz with a ground-based radar for LWP > 100. Also it would seem important to compare reflectivities too to document any statistical effect from liquid attenuation in these comparisons.

As shown in the updated Figure 5, when comparing at the same frequency, CPR exhibits mean Doppler velocity profiles similar to those of the surface radar regardless of LWP. Based on this, we conclude that the Doppler differences observed at NSA under high LWP conditions are mainly due to frequency-dependent scattering regime differences.

In addition, following the reviewer's suggestion, we have also included reflectivity profiles. While the reflectivity values are corrected for frequency difference, no correction for liquid attenuation is applied. Therefore, the differences seen in Fig. 5b are mainly attributable to liquid attenuation. This can introduce slight sampling differences between the two radars near the reflectivity threshold of -15 dBZ used for Doppler velocity, and we have added this point in the text:

(lines 385-387) "Additionally, stronger liquid attenuation at 94 GHz (see Fig. 5b) causes CPR to miss weak reflectivities that KAZR still detects, and this sampling difference may have slightly contributed to the mean Doppler velocity differences between the two radars (Fig. 5f)."

---

## Author Comment (AC2)

**Final author response to referee (egusphere-2025-2697)**

RC2: "Comment on egusphere-2025-2697", Anonymous Referee #2

**General comments**

This manuscript provides an evaluation of the EarthCARE Doppler product using data from two vertically pointing surface radars, one in the Arctic and the second in Antarctica. The EarthCARE Cloud Profiling Radar (CPR) is a spaceborne W-band radar that measures the W-band reflectivity of clouds and precipitation, like its predecessor, the NASA CloudSat radar. The larger antenna used on the EarthCARE CPR provides improved horizontal resolution and sensitivity; however, it also allows the measurement of Doppler velocity for the first time for a spaceborne atmospheric radar. Hence, validating the EarthCARE CPR velocity measurements is timely. The methodology for the evaluation makes use of a recently developed CPR simulator, which allows the surface-based radar data to be converted to CPR-like data with error bars. These results are then directly comparable with the CPR data. As noted, the work is timely and should be of interest to readers. I think the manuscript is well-written; the reported analysis is well-described. I am curious if the authors plan to extend this study to other situations, like convective systems. I have some minor comments below.

The authors sincerely thank the reviewer for the careful reading of our manuscript and for the constructive feedback. The reviewer's comments have been very helpful for refining our work.

Before providing detailed point-by-point responses to each comment, we would like to note one important update made during the revision process. To address comments raised by Reviewer #1, we extended the observation period June 2024 – February 2025 (~8 months) to June 2024 – July 2025 (~13 months). This extension required updating all figures and revising the associated text. Further detailed can be found in the revised manuscript.

**Specific comments**

**Comment #1**

Line 41 – the comparison with CloudSat indicates better resolution. Should this be better horizontal resolution? The vertical seems to be 500 m for both.

We agree with the reviewer that the vertical resolution of both spaceborne radars is about 500 m. To clarify, we have revised the sentence as follows:

(lines 39-42) "The EarthCARE CPR has higher sensitivity, better horizontal resolution, and reduced surface clutter contamination (Illingworth et al., 2015; Burns et al., 2016; Lamer et al., 2020) compared to the National Aeronautics and Space Administration (NASA) CloudSat CPR (Tanelli et al., 2008; Stephens et al., 2008, 2018)."

**Comment #2**

Sections 2.1 and 2.2.1 – these sections spend quite a bit of space on the sedimentation velocity. However, line 110 notes that "validation is conducted only from the perspective of the Doppler velocity best estimate." I'm not clear on this - does this mean that the comparisons are on the observed Doppler including both air motion and SVBE? If the study is just on the measured Doppler, then I'm puzzled by the space devoted to the sedimentation velocity. Please clarify. If sedimentation velocity is not really used in the comparison, I would shorten its mention to a couple of sentences.

We confirm that this study validates only the Doppler velocity. However, in the stratiform ice cloud cases considered here, where vertical air motion is typically weak, we assume that Doppler velocity biases are equivalent to sedimentation velocity biases, as described in lines 111-114 of the manuscript. Although sedimentation velocity is not directly part of the comparison presented, we included its description for context. This is because EarthCARE provides the first-ever global observations of sedimentation velocity, and this new capability will be of great interest to a much broader research community beyond the radar field. One of the central mission objectives is to enable the evaluation of fall speed parameterizations in weather and climate models. In this context, we considered it important to explicitly connect the Doppler velocities validated in this study to the sedimentation velocities that the wider community will use.

**Comment #3**

Line 130 – the horizontal velocity is assumed to be 9 m/s. Where does this come from?

The Orbital-Radar tool used in this study assumes a constant horizontal wind speed (default: 6 m s-1) throughout the whole atmosphere to perform the time – space conversion. To better represent high-latitude conditions, we updated the default to 9 m s-1 using radiosonde winds averaged below 10 km. However, over about one year of data, the daily mean wind speed exhibits large variability, and no single value can be considered optimal across the entire period and height range. Our assumption was that, given a sufficiently large sample, the mean vertical profiles of Z and V, as well as the Z-V relationship, would converge to climatological values, with random variability canceling out. This assumption was indeed supported by our results. When we extended the dataset from about 8 months to 13 months during the revision process, the mean Z and V profiles and Z-V relationships showed little change.

We added the following sentence in the revised manuscript.

(lines 164-165) "Even if this coordinate transformation is not optimal for every overpass, with a sufficiently large sample the measurements are expected to converge to climatological values."

**Comment #4**

Line 139 – I understand that we don't want to include more noise than necessary. However, I think maybe more detail would help convince the reader that the correct thermal and speckle noise levels are being used in the simulation.

In the revised manuscript, we have clarified the sentence as follows:

(lines 135-137) "For the Doppler velocity error, although the tool can reasonably estimate contributions from both satellite motion and receiver noise-related random error (i.e., thermal and speckle noise), our analysis incorporates only the satellite motion component to avoid overly noisy estimates."

**Comment #5**

Line 165 - can't things change quite a bit in three hours? I would think yes in the mid-latitudes but I'm not sure about the arctic. Is the difference in the times somehow included in the surface error bars?

We believe that our response to Comment #3 already addresses this concern.

**Comment #6**

Line 181 - are these offsets then applied to the surface-based radar data prior to running through the simulator?

Yes, these offsets are applied to the surface-based radar data prior to running it through the simulator. We revised the sentences as follows:

(lines 177-178) "As a result, calibration offsets of -2.1 dB for the NSA KAZR and -0.7 dB for the Neumayer FMCW radar were obtained, and these offsets were applied prior to processing with the Orbital-Radar tool."

**Comment #7**

Figure 2 and text starting around line 266 – I think there is a possibility of misinterpreting the bias correction. Specifically, the term "EarthCARE + e_p" could be interpreted as adding e_p to the EarthCARE observation, which would be the corrected EarthCARE. The caption does clarify that (b) and (f) are the EarthCARE before pointing correction. However, the authors might want to consider calling the uncorrected "EarthCARE" and the corrected "EarthCARE – bias" or EarthCARE – e_p".

We thank the reviewer for this suggestion. To avoid any possible misunderstanding, we have replaced "EarthCARE + e_p" with "EarthCARE (no e_p correction)" in Figure 2. The same update has also been applied to Figure 3.